# A BPTF Inhibitor That Interferes with the Multidrug Resistance Pump to Sensitize Murine Triple-Negative Breast Cancer Cells to Chemotherapy

**DOI:** 10.3390/ijms252111346

**Published:** 2024-10-22

**Authors:** Melanie M. Sinanian, Afshan Rahman, Ahmed M. Elshazly, Victoria Neely, Balaji Nagarajan, Glen E. Kellogg, April L. Risinger, David A. Gewirtz

**Affiliations:** 1Department of Pharmacology and Toxicology, Virginia Commonwealth University, Richmond, VA 23298, USA; melanie.sinanian@vcuhealth.org (M.M.S.); afshan.rahman@vcuhealth.org (A.R.); elshazlyam@vcu.edu (A.M.E.); 2Department of Pharmacology and Toxicology, Faculty of Pharmacy, Kafrelsheikh University, Kafrelsheikh 33516, Egypt; 3Philips Institute for Oral Health Research, School of Dentistry, Massey Cancer Center, Virginia Commonwealth University, Richmond, VA 23298, USA; neelyvl@vcu.edu; 4Department of Medicinal Chemistry, Virginia Commonwealth University, Richmond, VA 23298, USA; bnagarajan@vcu.edu (B.N.); gkellogg@vcu.edu (G.E.K.); 5Department of Pharmacology, University of Texas Health Science Center, San Antonio, TX 78229, USA; risingera@uthscsa.edu

**Keywords:** P-gp, P-glycoprotein, AU1, BPTF inhibitor, triple-negative breast cancer, efflux pump, multidrug resistance, multidrug resistance pump, breast cancer

## Abstract

Triple-negative breast cancer (TNBC) is associated with a generally poor prognosis due to its highly aggressive and metastatic nature, lack of targetable receptors, as well as the frequent development of resistance to chemotherapy. We previously reported that AU1, a small molecule developed as an inhibitor of BPTF (bromodomain PHD finger-containing transcription factor), was capable of sensitizing preclinical models of TNBC to chemotherapy in part via the promotion of autophagy. In studies reported here, we identify an additional property of this compound, specifically that sensitization is associated with the inhibition of the P-glycoprotein (P-gp) efflux pump. In silico molecular docking studies indicate that AU1 binds to active regions of the efflux pump in a manner consistent with the inhibition of the pump function. This work identifies a novel chemical structure that can influence multidrug efflux, an established mechanism of drug resistance in TNBC, that has not yet been successfully addressed by clinical efforts.

## 1. Introduction

Breast cancer is a heterogeneous disease that can be classified into molecular subtypes based on histologically determined markers, largely the expression status of the estrogen receptor (ER), progesterone receptor (PR), as well as human epidermal growth factor (HER2) [1,2,3]. Hormone-receptor-positive breast cancers account for the majority of breast cancer cases, where the expression of these receptors serves as the basis for targeted therapies and is thus generally associated with more positive clinical outcomes [4,5]. On the other hand, triple-negative breast cancer (TNBC), which does not express ER, PR, or HER2, has limited susceptibility to hormonal or targeted therapy [6,7]. Moreover, TNBC is also known for its more aggressive and metastatic nature and higher potential for disease recurrence [6,7]. Chemotherapy remains the standard of care for the treatment of TNBC following surgery and radiation [2,6,7,8]. PARP inhibitors, checkpoint inhibitors, and immunotherapies have been shown to have efficacy against distinct sub-populations of TNBC, demonstrating that targeted treatment strategies can be developed to improve therapy outcomes [7,9].

In recent years, the role of epigenetics in cancer development and progression has been widely recognized [10,11,12,13,14,15]. Epigenetics is broadly defined as the study of heritable changes to the genome that are not attributed to an altered DNA sequence (in contrast to genetic mutations). DNA accessibility for transcription, replication, and repair is controlled by epigenetic regulators, which include histone protein and post-translational modifications, DNA methylation, non-coding RNA, and enzymes such as chromatin remodeling complexes (CRCs) [16,17,18]. CRCs facilitate the movement of nucleosomes across the DNA, thereby allowing for control over gene expression, and are essential for processes such as growth and development as well as cancer-specific biology [16,18,19]. Given their putative involvement in the tumorigenesis and development of TNBC, recent attention has been directed towards epigenetic regulators as potential therapeutic targets [14,18,19]. 

The nucleosome remodeling factor (NURF) is a CRC and a member of the imitation switch (ISWI) family of ATP-dependent multidomain protein complexes [20,21]. As is the case with CRCs from other families, NURF has been implicated in human cancers [22,23,24,25,26,27]. NURF is comprised of an ATPase domain, SNF2L, a WD-repeat protein RbAP46/48, and a chromatin-binding protein, Bromodomain PHD-finger transcription factor (BPTF) [21]. BPTF is the largest subunit of NURF and is unique to and essential for its function [21,28]. This, along with bromodomains being considered “druggable”, makes BPTF particularly attractive as a therapeutic target. AU1 (rac-1), the compound that is the focus of this manuscript, is a commercially available small molecule inhibitor of BPTF that has a Kd = 2.8 µM for BPTF [28].

Chemotherapy treatment fails in 90% of cancer cases due to drug resistance [29]. It is, therefore, critical to address the phenomena of multidrug resistance (MDR) that occurs in all types of cancers. Multiple mechanisms of MDR have been identified in TNBC, including altered drug metabolism, altered drug targets, DNA repair, apoptosis inhibition, autophagy, increased drug efflux, and reduced drug influx [30,31,32,33]. In the context of MDR, drug efflux is facilitated by the overexpression of transporter proteins, resulting in the reduced exposure of cancer cells to antitumor drugs. ATP-binding cassette (ABC) transporters represent a large and diverse superfamily of ATP-dependent efflux proteins [34]. In TNBC, the overexpression of ABC transporters—most notably of canonical efflux pump P-glycoprotein (P-gp), multidrug resistance protein 1 (MRP1), and breast cancer resistance protein (BCRP)—has been demonstrated [31,35]. Critically, approximately 40% of TNBC cases show the overexpression of P-gp, making it a major obstacle in the successful treatment of cancer patients [36]. 

Over the course of many decades, numerous efflux pump inhibitor compounds have been developed and explored. The inhibition of efflux pumps such as P-gp can be achieved through competitive and noncompetitive inhibition as well as through other mechanisms such as disruption of cell membrane lipid integrity and ATP hydrolysis [34]. Four generations of P-gp inhibitors, categorized by metrics such as potency, selectivity, and drug interactions, have been developed and tested [34,37], but none have been approved for human use. The basis for the failure of these inhibitors in clinical trials has varied, including the necessity of utilizing intolerably high doses, high toxicity, drug interactions, and limited efficacies [34,37]. Nevertheless, efforts have continued to explore the use of natural and/or FDA-approved compounds that demonstrate P-gp inhibition activity for adjunctive therapeutic use with chemotherapy [38,39,40]. 

In the current work, we focus on P-gp of the ABC transporter family, one of the primary contributors to chemoresistance [37,41], and report that AU1, a compound developed as a BPTF inhibitor, enhances sensitivity to chemotherapy in TNBC by interfering with the function of the multidrug resistance pump.

## 2. Results

### 2.1. Assessment of Sensitization to the BPTF Inhibitor, AU1

A panel of FDA-approved drugs and compounds were added to genetically *BPTF*-silenced murine 4T1 TNBC cells, as characterized in the work by Tyutyunyk-Massey et al. [42], in a preliminary screen. Chemotherapeutic drugs that appeared to be more effective in cells with reduced BPTF expression were selected to be further assessed for activity with wildtype 4T1 and E0771-LMB murine TNBC cells in combination with the BPTF inhibitor AU1 since a pharmacological strategy would be the clinically relevant approach. Dose–response curves were performed in 4T1 and E0771-LMB murine TNBC cells to identify concentrations of AU1 that did not exhibit cytotoxicity alone. The nontoxic concentration of 2.5 µM AU1, which approximates its K_d_ of 2.8 µM [28], was selected to be used in combination with chemotherapies in subsequent experiments (Appendix A). 

Dose–response curves for various chemotherapy treatments with and without AU1 in the 4T1 and E0771-LMB cells are shown, respectively, in Figure 1 and Figure 2. For each agent, IC_50_ values were determined with and without AU1, and a fold change in IC_50_ values can be found for 4T1 cells in Table 1 and E0771-LMB cells in Table 2. The most pronounced overall IC_50_ fold changes for both 4T1 and E0771-LMB cell lines were observed for vinorelbine, which is a P-gp substrate. For 4T1 cells, the IC_50_ for vinorelbine alone is ~223 nM and shifts to ~27 nM when treated in combination with AU1, which translates to an 8-fold difference. Similarly, for E0771-LMB cells, the IC_50_ for vinorelbine shifts from 60 nM to 12 nM when treated with AU1, representing an approximately 5-fold difference. The addition of AU1 to 4T1 and E0771-LMB cell lines results in 2–5-fold increases in potency for vinblastine, vincristine, paclitaxel, and doxorubicin, which are all P-gp substrates. Conversely, AU1 does not appear to sensitize cells to either 5-FU or cisplatin, drugs that are generally not considered to be substrates for the P-gp efflux pump [43]. These data led to the hypothesis that AU1 sensitizes these cells preferentially to drugs that are P-gp substrates. 

Consistent with this hypothesis, when 4T1 cells were treated with gemcitabine (Appendix A), drug sensitivity was actually reduced instead of increased. This observation is similar to findings generated by Bergman et al. [44], which showed increased sensitivity to gemcitabine in P-gp overexpressing cell lines. 

Dose–response curves for vinorelbine, vinblastine, and doxorubicin with and without AU1 administration were also performed with human MDA-MB-231 TNBC cells, as shown in Figure 3. Fold changes in IC_50_ values are presented in Table 3. The IC_50_ shifts are quite modest, suggesting that AU1 does not sensitize the MDA-MB-231 cells to the chemotherapeutic agents, in contrast to the findings with the 4T1 and E0771-LMB cell lines. 

Clonogenic survival assays were performed to further confirm the sensitization conferred by AU1 in combination treatments. Figure 4 shows the results from clonogenic survival assays performed with the select chemotherapies explored in Figure 1 in 4T1 cells. Utilizing drug concentrations that alone did not significantly suppress colony formation, colony formation is virtually eliminated in combination with AU1 for vinorelbine, vinblastine, and vincristine and significantly suppressed for paclitaxel and doxorubicin. 

### 2.2. Growth Arrest and Cell Death for the Combination of Chemotherapy with the BPTF Inhibitor, AU1

In order to further establish the durability of the growth suppression induced by the combination of AU1 with select chemotherapeutic agents, temporal response studies were performed for the combination of either vinorelbine or vinblastine with AU1 in 4T1 cells, as is shown in Figure 5A,B, respectively. Cell growth was largely unaffected with either chemotherapy or AU1 alone, whereas the combination treatments resulted in complete suppression of proliferative capacity for 16–18 days, as shown in the expanded dotted regions. This held true for both combination treatment protocols, specifically where AU1 treatment was continued throughout the duration of the study, as well as the regimen where AU1 was removed after the fourth day when the chemotherapy was also removed. 

With the combination treatment, the cells demonstrated classical traits of senescence, such as flattened morphology. The extent of senescence on days three and four is shown in Appendix A for the vinorelbine and AU1 combination. Furthermore, analyses by flow cytometry on day four of treatment indicated that combination treatment of AU1 with vinorelbine, vincristine, vinblastine, paclitaxel, or doxorubicin increased apoptosis of 4T1 cells as compared to each of the chemotherapeutics alone (Figure 6). Consequently, both senescence and apoptosis appear to contribute to the sustained growth arrest of the combination treatment strategy.

### 2.3. Evidence That AU1 Is Acting to Inhibit the Multidrug Resistance Pump

While AU1 was developed as a BPTF inhibitor, it became evident that the drugs to which both 4T1 and E0771-LMB cells were sensitized when treated in combination with AU1 (Table 1 and Table 2) were known substrates of efflux pumps such as P-gp and MRP1 [44]. Classes of drugs that did not exhibit sensitization when treated in combination with AU1 were from platinum-based and antimetabolite classes, both of which are not substrates of efflux proteins MRP and P-gp [45]. Additionally, this sensitization was not observed in the MDA-MB-231 cell line. This indicated that AU1 could be acting to modulate efflux pump(s). Figure 7D indicates that the P-gp multidrug resistance pump is expressed in both the 4T1 and the E0771-LMB cells, which confers resistance to the microtubule targeting and anthracycline drug classes [45], consistent with our findings that AU1-sensitized P-gp-expressing lines to these agents; however, this protein is not detectable in the MDA-MB-231 cells. We found that administration of AU1 did not appear to affect P-gp protein expression in either the 4T1 or E0771 cells, suggesting that any modulation of P-gp activity was likely to be post-translational (Appendix A). 

The ability of AU1 to act as an efflux pump inhibitor (for MRP1 and/or P-gp) was further investigated using a calcein-AM-based kit multidrug resistance efflux kit. Briefly, the assay employs calcein-AM, which is a nonfluorescent dye that passively enters cells and is cleaved by intracellular esterases to form the fluorescent product calcein. Calcein is a substrate of P-gp and MRP efflux pumps, and its presence in cells can be quantified via flow cytometry. Cells with high expression of and/or uninhibited efflux pump proteins will efflux more calcein and emit lower fluorescence, while cells that have inhibited efflux pumps and/or lowered expression will retain calcein and emit higher fluorescence.

Representative histograms, as well as mean fluorescent intensity values, averaged across multiple biological replicates for DMSO control and AU1-treated cells, are shown in Figure 7. 4T1 and E0771-LMB cell lines, in Figure 7A,B, respectively, demonstrate that AU1 significantly inhibits the efflux of calcein-AM from both 4T1 and E0771-LMB cells, whereas administration of AU1 to MDA-MB-231 cells had no statistical effect on calcein-AM retention (Figure 7C). The more modest inhibition in the E0771-LMB line as compared to 4T1 cells is consistent with the lower levels of P-gp in the E0771-LMB cells, as shown in Figure 7D. 

We next compared the sensitizing effects of AU1 with a classical inhibitor of drug efflux, verapamil, which is clinically employed as a calcium channel blocker used to treat cardiac arrhythmias, hypertension, and angina and is a known inhibitor of P-gp [34,46]. 4T1 cells subjected to ranges of verapamil concentrations up to 5 µM appeared to grow similarly to controls, indicating that even high concentrations of verapamil had no apparent cytotoxic effect (Appendix A). Concentrations of 0.5–5 µM verapamil promoted similar sensitization to vinorelbine as that observed for AU1 (Figure 8A).

Temporal response studies of these drug combinations also show similar growth suppression for AU1 and either 2.5 or 5 µM of verapamil; in both cases, recovery of proliferative capacity was observed around day 16 (Figure 8B). Extended time-based viability profiles exploring different treatment regimens of AU1 or verapamil with vinorelbine, shown in Figure 8C, demonstrated that when removing AU1/verapamil on day 4, a similar growth suppression and return of proliferative capacity resulted as compared to the continued treatment. Taken together, these data in Figure 8B,C suggest that combination treatments of vinorelbine with AU1 or verapamil may be acting through similar mechanisms. 

In a clonogenic survival assay, shown in Figure 8D, verapamil alone had no effect on colony growth, consistent with the data presented in Appendix A, whereas administration of 50 nM vinorelbine yielded a modest reduction in the number of colonies. However, the combination treatments of either AU1 or verapamil with vinorelbine resulted in complete suppression of colony growth. Annexin V/PI flow cytometry revealed that apoptosis was a primary mechanism of cell death (Figure 8E) for the combination of vinorelbine and verapamil. 

To further support the premise that the AU1 was acting similarly to verapamil, 4T1 cells were treated with nontoxic concentrations of both verapamil and AU1 (Figure 9A) along with vinorelbine (Figure 9B). The AU1 and verapamil combination treatments with vinorelbine were no more effective in suppressing cell growth than either AU1 or verapamil with vinorelbine. These data are consistent with verapamil and AU1 acting thorough a similar mechanism to sensitize cells P-gp expressing cells to vinorelbine.

### 2.4. Molecular Docking of AU1 Binding to MDRP

Finally, to further establish the interaction of AU1 with the multidrug resistance pump, we performed a computational study of AU1 with the multidrug resistance pump (MDRP) target protein. The GOLD [47] molecular docking suite was used to perform the interaction study; the results indicate that the computational affinity and specificity of AU1 for MDRP is high, and a number of key amino acid interactions can be identified. 

The top-ranked structure of AU1 bound to the CryoEM structure of MDRP, with a GOLD score of 83.57, was subjected to an interaction analysis. Figure 10 shows the summary of all interactions: AU1 identified 10 amino acid residues within the protein’s binding pocket that contributed to bonded and non-bonded interactions. AU1 makes strong direct hydrogen bond interactions with the amino acid residues GLN990 and TRP232 (shown in magenta sticks in Figure 10B and green color in Figure 10D) and also interacts with many hydrophobic residues through non-bonded interactions in the binding pocket, taking a semi-L to semi-U shape compared to the more globular bound vincristine (Appendix A) [48]. Interestingly, AU1 identified four similar residues to those seen with vincristine in the CryoEM structure, namely PHE343, GLN347, MET949, and GLN990. In both, GLN990 makes a direct hydrogen bond, while the other residues contribute through non-bonded interactions (Figure 10D). AU1 identified an additional amino acid residue, TRP232, with a stabilizing hydrogen bond that could potentially increase the affinity of AU1 to MDRP, increasing inhibition. In comparing the AU1 RMSDs of the top three ranked poses, the deviation values ranged between 0.10 Å and 0.22 Å, which further confirms that the interaction is more specific. 

This proposed high affinity and specificity binding of AU1 in the active region of MDRP could potentially lead to observable inhibition activity, and our results are in agreement with those of the earlier published structural studies [48,49]. 

## 3. Discussion 

Due to its aggressive, metastatic nature and the lack of targetable receptors, TNBC remains the most lethal breast cancer subtype and poses a particular challenge for treatment options. Cytotoxic chemotherapy remains the standard of care for TNBC, with immunotherapy and antibody-drug conjugates taking on more prominent roles in recent years [7,9]. However, drug resistance remains one of the primary reasons for chemotherapy treatment failure [31]. Of all the subtypes of breast cancer, TNBC is associated with the highest expression of efflux pumps, with 40% of TNBC tumors demonstrating the overexpression of P-gp [36]. 

In the current work, we initially sought to employ AU1 as a pharmacological inhibitor of BPTF in a continued effort to determine how NURF impacts the sensitivity of breast cancer to several different chemotherapeutic agents. We showed that AU1 administration conferred sensitization of both 4T1 and E0771-LMB murine TNBC cells to chemotherapies that are employed to treat TNBC, including agents from the taxane, vinca alkaloid, and anthracycline drug classes. These findings are consistent with previous work from our laboratory where 4T1 cells were sensitized to doxorubicin (as well as other topoisomerase II poisons) through both genetic inhibition of BPTF or treatment with AU1 [42]. In the current work, we further demonstrate that AU1 also functions as an efflux pump inhibitor. 

The combination treatment with AU1 was highly efficacious in both 4T1 and E0771-LMB triple-negative mammary cancer cells. Our studies demonstrated that colony formation could be dramatically suppressed. Furthermore, in the studies with both vinorelbine and vinblastine, the growth suppression by the combination treatment with AU1 was shown to be durable, lasting until proliferative recovery began to occur around day 16. Importantly, this sustained suspension of growth occurred both with continuous AU1 treatment as well as when AU1 and chemotherapy treatment were terminated on day 4. Both apoptosis and senescence appeared to contribute to the growth suppression observed by the combination treatment strategy. The proliferative recovery from growth arrest is a consistent feature of radiation and/or chemotherapy-treated cells that have undergone senescence, as shown extensively by us [50,51,52,53] as well as other laboratories [54,55,56,57]. 

Our studies strongly support the conclusion that AU1 is sensitizing the triple-negative mammary cancer cells to chemotherapy via inhibition of the P-gp efflux pump. In this context, we demonstrated the expression of the pump protein in the 4T1 cells and in the E0771-LMB cells and its absence in the MDA-MB-231 cells. These findings were further supported by the calcein-AM-based multidrug resistance kit experiments, where we showed that both triple-negative murine cell lines appeared to efflux calcein, a substrate of P-gp, and where administration of AU1 inhibited this efflux mechanism. In the MDA-MB-231 cells, calcein efflux remained minimal and similar between all samples, consistent with low efflux pump expression. 

The sensitization of 4T1 and E0771-LMB cells to vinorelbine, vincristine, vinblastine, paclitaxel, and doxorubicin was notable as these are all well-established efflux pump substrates [37]. The different degrees of sensitization may be associated with the differential affinity of these agents for the pump. In contrast, we observed a lack of sensitization to 5-FU and cisplatin, drugs that lack the large and lipophilic properties associated with high-affinity efflux substrates [58,59]. The reduced sensitivity to gemcitabine upon treatment in combination with AU1 was initially confusing; however, a previous publication also showed similar results, with five different cell lines showing reduced rather than enhanced sensitivity to gemcitabine upon treatment with verapamil, a known efflux pump inhibitor. The proposed basis for this finding suggested an interplay between efflux pump activity and transient regulation in the activity of deoxycytidine kinase, which is an enzyme that contributes to the activation of gemcitabine [44]. Further support for these findings is derived from the observation that AU1 does not sensitize MDA-MB-231 cells, which lack the efflux pump to these agents.

We further compared the outcomes of the chemotherapy combination treatments with either AU1 or verapamil, where we were able to identify similarities between the response of 4T1 cells to the treatments. The dose–response curves yielded similar IC_50_ shifts, and clonogenic survival showed the same impressive growth suppression. Furthermore, the time-based viability assay showed identical growth suppression, and the mechanism of growth suppression was also shown to be apoptosis. By combining AU1 and verapamil with chemotherapy treatment, we established that there was no significant increase in sensitization, indicating that the mechanisms of action of the AU1 compound and verapamil were likely similar. Finally, our molecular docking studies provided clear evidence of AU1 being associated with the pump protein with a high GOLD score and an additional hydrogen bond interacting residue in the binding pocket.

As murine cell lines were employed to establish the capacity of AU1 to act as an efflux pump inhibitor in this work, further studies employing human TNBC cell lines that express P-gp could be performed to validate the use of AU1 or similar compounds. However, as shown in studies by Jain et al. [60], P-gp is highly conserved among human and mouse species, demonstrating similar binding sites, and therefore findings in murine tumor cells are likely to be relevant to human triple-negative breast cancer cells.

We acknowledge certain limitations with the current work and recognize additional experiments that could contribute to the rigor of the current findings. For instance, we cannot conclude with certainty that AU1 and verapamil are acting in an identical fashion to interfere with drug efflux. Knockdown of the pump in the 4T1 (or E0771-LMB) cells should theoretically largely eliminate the observed sensitization. Conversely, the overexpression of P-gp in the MDA-MB-231 cells should serve to make these cells responsive to chemosensitization by AU1. Nevertheless, taken together, our studies are fully consistent with the conclusion that AU1 can enhance sensitivity to select antitumor drugs by interfering with the P-gp efflux pump.

## 4. Materials and Methods

### 4.1. Cell Lines

Wildtype 4T1 (gift from Fred Miller, Wayne State University), E0771-LMB (gift from Joseph Landry, Virginia Commonwealth University), and MDA-MB-231 (ATCC, catalog no. HTB-26) cells were cultured in complete media (DMEM (11995-065, Gibco; Waltham, MA, USA) supplemented with 10% (*v*/*v*) fetal bovine serum (FBS) (SH30066.03, Thermo Scientific; Waltham, MA, USA) and 100 U/mL penicillin G sodium and 100 μg/mL streptomycin sulfate (15140122, Gibco). Cells were used within 20 passages and were tested and confirmed negative for *mycoplasma* contamination bimonthly using the MycoStrip™ 50 Kit (rep-mysnc-50, InvivoGen; San Diego, CA, USA).

### 4.2. Drug Treatments 

AU1 (synthesized and provided by Dr. William Pomerantz, University of Minnesota), vinblastine (S4505, SelleckChem; Houston, TX, USA) vincristine (S1241, SelleckChem; Houston, TX, USA), 5-FU (3257, Tocris; Minneapolis, MN, USA) verapamil (HY-A0064, MedChemExpress; Monmouth Junction, NJ, USA), and paclitaxel (NDC code 70860-200-05, Athenex; Buffalo, NY, USA) were dissolved in DMSO. Vinorelbine (V2264, Sigma-Aldrich; St. Louis, MO, USA), doxorubicin (D1515, Sigma-Aldrich; St. Louis, MO, USA), and cisplatin (2251, Tocris; Minneapolis, MN, USA) were dissolved in sterile distilled water or media. 

The tumor cells were allowed to adhere for 4–6 h after plating. AU1 and combination treatment groups were then pre-treated overnight with AU1 for at least 12 h. Cells were treated the following day with chemotherapeutic drugs with or without the simultaneous administration of AU1 for 96 h. Experiments with endpoints beyond 96 h were either continued in AU1-treated or untreated media, as indicated. Drug- and vehicle-treated media changes were performed every two days for experiments performed in 6-well plates. 

### 4.3. Dose–Response Assessments

Cells were plated at suitable densities (1500 cells/well for 4T1 cells and 3000 cells/well for E0771-LMB and MDA-MB-231 cells) in 96-well plates, allowed to adhere to the plates for 4–6 h, pre-treated with AU1 overnight (at least 12 h), and treated the following day with serially diluted chemotherapies with and without AU1 for 96 h. Cells were evaluated via the MTS viability assay. 

### 4.4. Clonogenic Survival Assay

4T1 cells plated at 300 cells/well in 6-well plates were allowed to adhere for 4–6 h, pre-treated with AU1 overnight as appropriate (at least 12 h), and treated the following day with chemotherapy, AU1, or combination treatments for 96 h. Chemotherapy drug treatment was then terminated, and AU1 administration continued, as indicated, until ~144 h, when colonies began to converge. Plates were fixed with methanol, stained with crystal violet, and manually counted. Counts were normalized with respect to the control treatment group. 

### 4.5. Temporal Response Assay

Cells were plated at 3000 cells/well in 6-well plates, allowed to adhere for 4–6 h, pre-treated with AU1 overnight as appropriate (at least 12 h), and treated the following day with chemotherapy, AU1, or combination treatments for 96 h. Drug treatment was then terminated, and AU1 administration continued or discontinued until day 20. Cells were trypsinized, stained with 0.4% trypan blue (T01282, Sigma-Aldrich), and counted on the indicated days using a hemocytometer; growth curves were generated from the collected data. 

### 4.6. Determination of Apoptosis

Cells were plated at suitable densities, allowed to adhere, and pre-treated with AU1 overnight, as appropriate. The following day, cells were treated with the respective agents at the indicated concentrations for 96 h. The extent of apoptotic cell death was measured using Annexin V-FITC/propidium iodide staining. Cells were trypsinized, washed with 1X PBS, and stained according to the manufacturer’s protocol (556547, Annexin V-FITC Apoptosis Detection Kit; BD Biosciences; Franklin Lakes, NJ, USA). Fluorescence was quantified via flow cytometry using BD FACS Canto II (v8.0.1, BD Biosciences; Franklin Lakes, NJ, USA) and BD FACS Diva software (v10.8.1, BD Biosciences) at the Flow Cytometry Core Facility at Virginia Commonwealth University. For all flow cytometry experiments, 10,000 cells per replicate were analyzed, and three replicates for each condition were analyzed per independent experiment unless otherwise stated. All experiments were performed with cells protected from light.

### 4.7. Western Blot Analysis

Western blotting was performed as previously described [16]. Briefly, after the indicated treatments, cells were harvested, pellets were lysed, and protein concentrations were determined via the Bradford Assay (5000205, Bio-Rad Laboratories; Hercules, CA, USA). Protein samples were loaded and subjected to SDS-polyacrylamide gel electrophoresis, transferred to polyvinylidene difluoride membrane, and blocked with 5% BSA in 1X PBS with 0.1% Tween 20 (BP337, Fisher; Hampton, NH, USA). The membrane was incubated overnight at 4 °C with the indicated primary antibodies at a dilution of 1:500 for P-gp (#A19093, ABclonal; Woburn, MA, USA) and 1:4000 for β-actin (#4970, Cell Signaling Technology; Danvers, MA, USA) with 5% BSA in 1X PBS. The membrane was then washed, horseradish peroxidase (HRP)-conjugated anti-rabbit (#7074, Cell Signaling Technology) secondary antibody was added at a dilution of 1:2000 with 5% BSA in 1X PBS for 2 h at room temperature, and the membrane was washed three times in 1X PBS with 0.1% Tween 20. Blots were developed using Pierce-enhanced chemiluminescence reagents (32132, Thermo Scientific) on the Bio-Rad ChemiDoc System. Image-J software (1.53c, National Institutes of Health; Bethesda, MD, USA) was utilized for the quantification of Western blots. 

### 4.8. Multidrug Resistance Assay 

Efflux pump activity and inhibition were assessed using the calcein-AM-based MDR Assay Kit (600370, Cayman; Ann Arbor, MI, USA). Cells were plated at a density of 150,000 cells/well in 6-well plates and allowed to adhere for 4–6 h. AU1 was administered overnight, for at least 12 h before running the assay. Administration of control compounds, verapamil, and cyclosporin A, as well as calcein-AM and PI were performed according to the manufacturer’s protocol. Cells were collected and analyzed using BD FACSCanto II and BD FACS Diva software. For all flow cytometry experiments, 10,000 cells per replicate were analyzed, and three replicates for each condition were analyzed per independent experiment. All experiments were performed with cells protected from light. Flow cytometry data were analyzed using Flowjo software (v9.9; Tree star Inc.; Ashland, OR, USA).

### 4.9. MTS Assays

The MTS assay was performed according to the manufacturer’s protocol (AB197010, Abcam; Cambridge, UK) and absorbance, which correlates to the number of viable cells per well, and recorded at 490 nm on a Biotek ELX800 Universal Microplate Reader (Winooski, VT, USA). 

### 4.10. Statistics

Unless otherwise indicated, all quantitative data are shown as the mean ± SEM from at least three independent experiments (biological replicates), all of which were performed in triplicate or duplicates (technical replicates). GraphPad Prism 9.0 software was used for statistical analysis. All data were analyzed using either a one- or two-way ANOVA, as appropriate, with Tukey or Sidak post hoc. IC_50_ values were calculated using dose–response assessment fittings with four-parameter logistic regressions.

### 4.11. Molecular Docking 

GOLD v5.6 (from Cambridge Crystallographic Data Center, Cambridge, UK) [47] was employed to study the interactions between the AU1 small molecule ligand and the MDRP protein. The three-dimensional structure of MDRP was obtained from the published CryoEM structure (PDBID: 7A69) [48]. Protein and ligand preparations were carried out using SYBYL X2.1 (Tripos Associates, St. Louis, MO, USA), which included the addition of hydrogen and missing atoms, protonation of residues, removal of steric clashes, and energy minimization. The centroid of the bound ligand from the CryoEM structure was taken as the center, and a 16-Å radius was defined as the binding site from this centroid. Docking was performed for 100 genetic algorithm runs with 100,000 iterations, and the early termination option was disabled. The GOLD fitness score was calculated from the contributions of hydrogen bonds and van der Waals interactions between the protein and ligand [47]. From the GOLD-based docking, the best-sampled pose (highest GOLD score) was selected and analyzed for interactions. Discovery Studio visualizer was used to make 2D interaction profiles (BioVia, D.S (2019) Accelrys Software, Cambridge, UK, https://discover.3ds.com, accessed on 20 December 2023), and PyMOL [61] was used to illustrate the 3D interactions. 

## Figures and Tables

**Figure 1 ijms-25-11346-f001:**
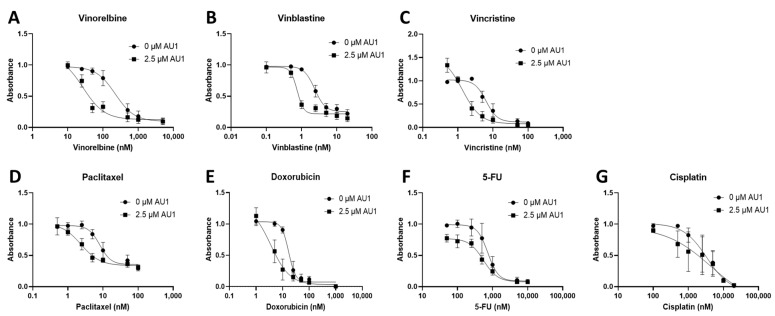
Dose–response assessments for 4T1 cells treated with select chemotherapeutic agents with and without AU1 in combination. Dose–response curves of serially diluted chemotherapeutic drugs (**A**) vinorelbine, (**B**) vinblastine, (**C**) vincristine, (**D**) paclitaxel, and (**E**) doxorubicin, (**F**) 5-FU, (**G**) and cisplatin performed with and without 2.5 µM AU1. Corresponding IC_50_ values and calculated fold change are listed in Table 1. Plated 4T1 cells were allowed to adhere to 96-well plates, pre-treated with AU1 overnight as appropriate, and treated the following day with serially diluted chemotherapies with and without AU1 for 96 h. Cells were evaluated via the MTS viability assay and absorbance values were normalized to DMSO controls. Results are means ± SEM of at least three independent experiments.

**Figure 2 ijms-25-11346-f002:**
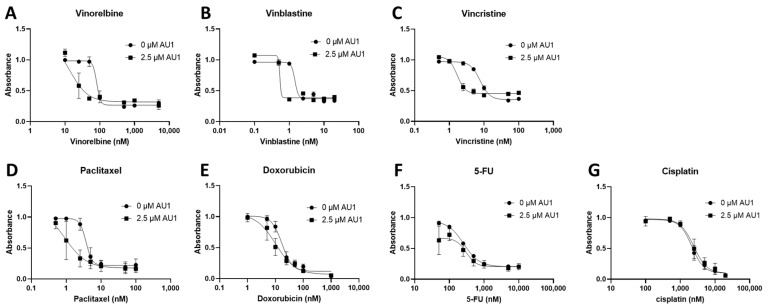
Dose–response assessments for E0771-LMB cells treated with select chemotherapeutic agents without and with AU1 in combination. Dose–response curves of serially diluted chemotherapies (**A**) vinorelbine, (**B**) vinblastine, (**C**) vincristine, (**D**) paclitaxel, and (**E**) doxorubicin, (**F**) 5-FU, (**G**) and cisplatin performed with and without 2.5 µM AU1. Corresponding IC_50_ values and calculated fold change are listed in Table 1. E0771-LMB cells were allowed to adhere to 96-well plates, pre-treated with AU1 overnight as appropriate, and dosed the following day with serially diluted chemotherapies with and without AU1 for 96 h. Cells were evaluated via the MTS viability assay and absorbance values were normalized to DMSO controls. Results are means ± SEM of at least three independent experiments.

**Figure 3 ijms-25-11346-f003:**
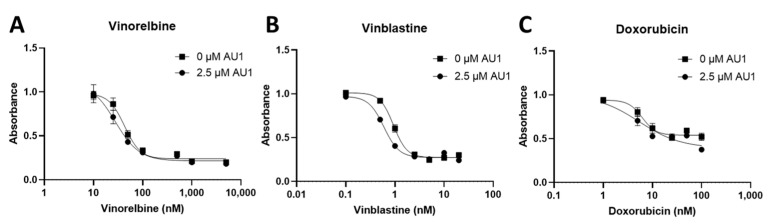
Dose–response assessments of MDA-MB-231 cells treated with select chemotherapies without and with AU1 in combination. Dose–response curves of serially diluted chemotherapies (**A**) vinorelbine, (**B**) vinblastine, and (**C**) doxorubicin performed without and with 2.5 µM AU1. Corresponding IC_50_ values and calculated fold change are listed in Table 3. MDA-MB-231 cells were allowed to adhere to 96-well plates, pre-treated with AU1 overnight as appropriate, and dosed the following day with serially diluted chemotherapies with and without AU1 for 96 h. Cells were evaluated via the MTS viability assay and absorbance values were normalized to DMSO controls. Results are means ± SEM of at least three independent experiments.

**Figure 4 ijms-25-11346-f004:**
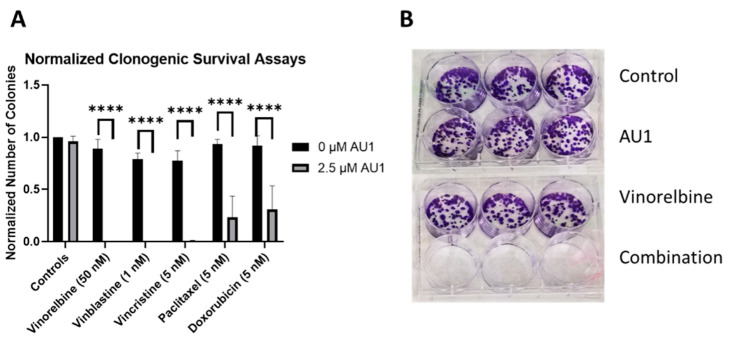
Sensitization of 4T1 cells to various chemotherapeutic drugs by AU1. (**A**) Clonogenic survival assays of select chemotherapies, where colony counts were normalized to their respective controls. **** *p* ≤ 0.0001 by one-way ANOVA with Dunnett post hoc. (**B**) Representative image of clonogenic survival assay results with vinorelbine and AU1. Plated 4T1 cells were allowed to adhere overnight, treated with chemotherapy, AU1, or the combination for 96 h. Drug treatment was then terminated and AU1 administration continued, as appropriate, until ~144 h when colonies began to converge. Plates were fixed, stained with crystal violet, and manually counted. Results are means ± SEM of at least three independent experiments.

**Figure 5 ijms-25-11346-f005:**
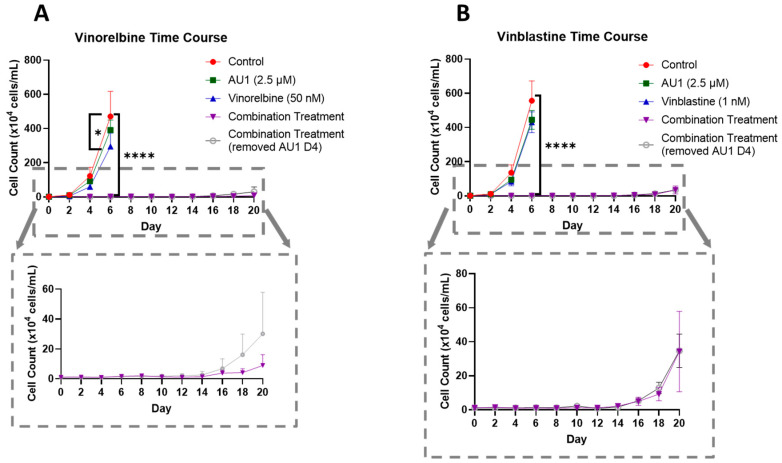
Growth curves of 4T1 cells treated with vinca alkaloids alone or in combination with AU1. (**A**) Vinorelbine with AU1 and (**B**) vinblastine with AU1 growth curves performed with trypan blue exclusion. Expanded figure, as indicated by the dashed lines, show only the combination treatment. * *p* ≤ 0.05 and **** *p* ≤ 0.0001 compared to controls by one-way ANOVA with Dunnet’s post hoc. Results are means ± SEM of at least three independent experiments.

**Figure 6 ijms-25-11346-f006:**
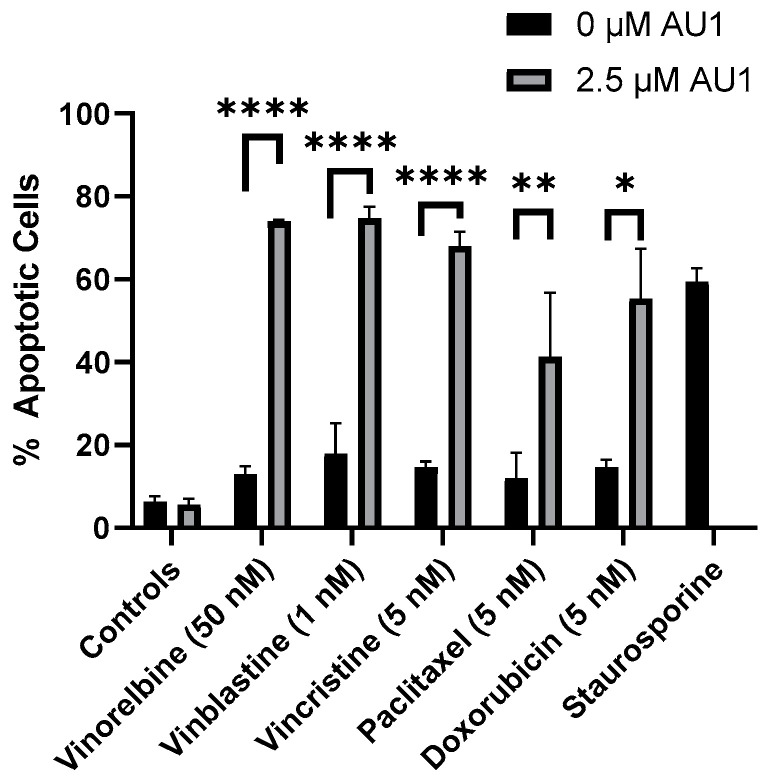
Quantification of apoptosis in 4T1 cells treated with various chemotherapeutic agents alone or in combination with 2.5 µM AU1. Apoptosis was measured with FITC Annexin/PI for both vinorelbine, vinblastine, vincristine, or paclitaxel alone or in combination with 2.5 µM AU1 after 96 h of treatment. Cells treated with 50 nM staurosporine for 24 h are included as a positive control for apoptosis. * *p* ≤ 0.05, ** *p* ≤ 0.01, and **** *p* ≤ 0.0001 compared to controls by one-way ANOVA with Tukey post hoc. Results are means ± SEM of at least three independent experiments.

**Figure 7 ijms-25-11346-f007:**
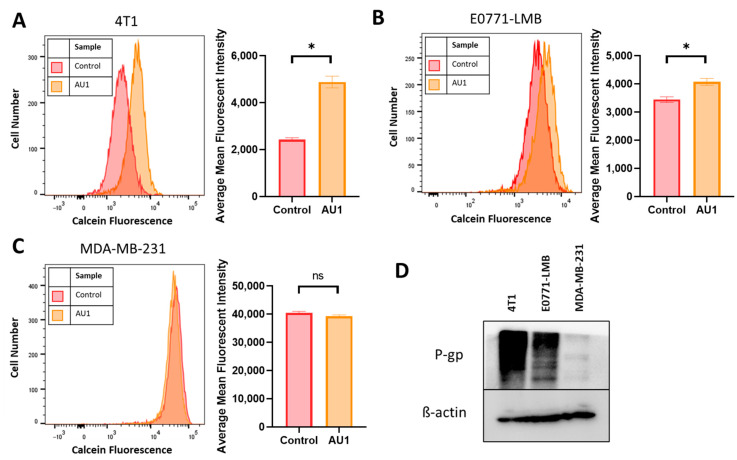
Efflux pump inhibition of AU1 and P-glycoprotein (P-gp) expression in 4T1 and E0771-LMB cells. Histograms generated from multidrug-resistance assay kit that evaluate for efflux pump inhibition activity of AU1 based on calcein-AM retention in (**A**) 4T1 cells, (**B**) E0771-LMB cells, and (**C**) MDA-MB-231 cells. (**D**) Western blotting for P-gp efflux pump for each cell line. * *p* ≤ 0.05 and ns (not significant) compared to controls by unpaired student *t*-tests. All histograms and blots are taken as representative results from 3 biological replicates. Averaged results are means ± SEM of at least three independent experiments.

**Figure 8 ijms-25-11346-f008:**
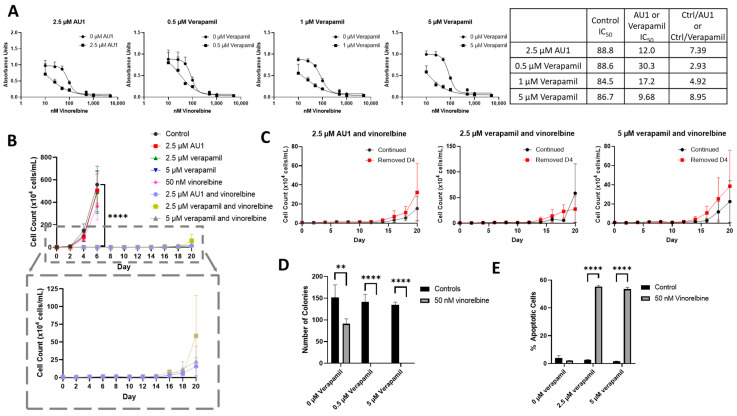
Sensitization of 4T1 cells to vinorelbine by verapamil. (**A**) Dose–response curves of serially diluted vinorelbine with 0.5, 1, and 5 µM verapamil and 2.5 µM AU1, with corresponding IC_50_ values, shifts, and fold changes normalized to DMSO control. (**B**) Growth curves of vinorelbine with 2.5 µM AU1, or 2.5 or 5 µM verapamil. Expanded figure, as indicated by the dashed lines, shows only the combination treatments with continued AU1/verapamil administration. **** *p* ≤ 0.0001 compared to control by one-way ANOVA with Dunnet post hoc. (**C**) Combination treatment growth curves with discontinuation of verapamil on day 4 or continuous administration of AU1/verapamil. (**D**) Clonogenic survival assay of cells treated with vinorelbine and 0.5 µM or 5 µM verapamil. ** *p* ≤ 0.01 and **** *p* ≤ 0.0001 compared to controls by one-way ANOVA with Dunnet post hoc (**E**) Apoptosis measured with FITC Annexin/PI for vinorelbine alone or in combination with 2.5 µM or 5 µM verapamil after 96 h of treatment. and **** *p* ≤ 0.0001 compared to controls by one-way ANOVA with Dunnet post hoc All results are means ± SEM of at least three independent experiments.

**Figure 9 ijms-25-11346-f009:**
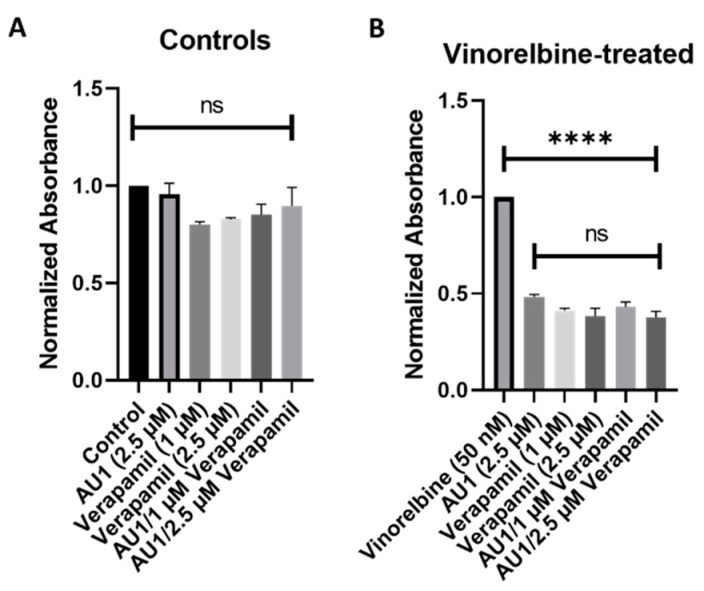
Influence of the combination of AU1 and verapamil on sensitivity to vinorelbine in 4T1 cells. 4T1 cells treated with AU1, 1 or 2.5 µM verapamil, or the combinations of AU1 and verapamil simultaneously either (**A**) alone or (**B**) in combination with 50 nM vinorelbine. Cells were pre-treated with AU1 and/or verapamil and treated the following day with AU1 and/or verapamil (**A**) without and (**B**) with vinorelbine for 96 h. Cell viability was evaluated via the MTS viability assay and normalized to (**A**) DMSO and (**B**) vinorelbine controls. **** *p* ≤ 0.0001 and ns (not significant) compared to controls by one-way ANOVA with Tukey post hoc. Results are means ± SEM of at least three independent experiments.

**Figure 10 ijms-25-11346-f010:**
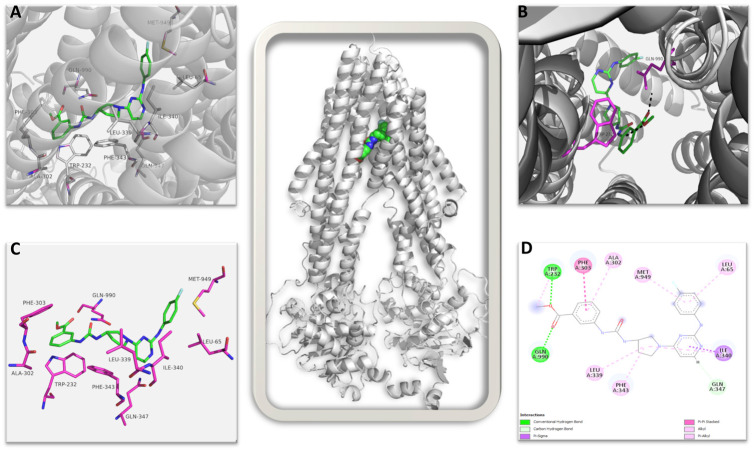
Multidrug resistance pump (MDRP): AU1 complex. The image in the center shows the complete 3D structure of MDRP in cartoon representation (in grey color) with the bound AU1 shown in spheres. (**A**) Shows the ligand AU1 in the binding pocket in green stick with interacting amino acid shown as labeled magenta sticks. (**B**) The amino acid residues contributing to direct hydrogen bond interaction to AU1 are shown. (**C**) AU1 shown in green sticks with interacting amino acid residues in magenta sticks in the absence of secondary structures. (**D**) 2D representation of all possible interactions of AU1 to MDRP is shown. Each type of possible interactions is shown in respective colors, e.g., green: direct hydrogen bond, etc.

**Table 1 ijms-25-11346-t001:** IC_50_ values from dose–response curves of 4T1 cells in Figure 1.

IC_50_ Values	Control	2.5 µM AU1	Control/AU1
Vinorelbine (nM)	223 ± 60	27.0 ± 16.7	8.29
Vinblastine (nM)	2.34 ± 0.26	0.75 ± 0.10	3.14
Vincristine (nM)	6.4 ± 0.8	1.34 ± 0.59	4.72
Paclitaxel (nM)	8.13 ± 1.34	2.37 ± 1.11	3.43
Doxorubicin (nM)	16.9 ± 2.0	3.6 ± 3.5	4.71
5-FU (nM)	737 ± 125	551 ± 87	1.34
Cisplatin (µM)	2.79 ± 0.55	2.40 ± 0.65	1.16

**Table 2 ijms-25-11346-t002:** IC_50_ values from dose–response curves of E0771-LMB cells in Figure 2.

IC_50_ Values	Control	2.5 µM AU1	Control/AU1
Vinorelbine (nM)	59.8 ± 9.4	11.7 ± 42.0	5.11
Vinblastine (nM)	1.46 ± 0.43	0.538 ± 0.537	2.7
Vincristine (nM)	7.75 ± 0.41	1.68 ± 0.16	4.26
Paclitaxel (nM)	3.71 ± 0.43	0.95 ± 1.5	3.91
Doxorubicin (nM)	17.3 ± 2.1	10.4 ± 3.8	1.66
5-FU (nM)	259 ± 22	182 ± 92	0.85
Cisplatin (µM)	2.15 ± 0.24	2.42 ± 0.42	0.89

**Table 3 ijms-25-11346-t003:** IC_50_ values from dose–response curves of MDA-MB-231 cells in Figure 3.

IC_50_ Values	0 µM AU1	2.5 µM AU1	Control/AU1
Vinorelbine (nM)	42.9 ± 3.8	27.8 ± 7.1	1.54
Vinblastine (nM)	0.93 ± 0.035	0.596 ± 0.027	1.57
Doxorubicin (nM)	6.23 ± 0.91	5.35 ± 3.57	1.16

## Data Availability

Data is contained within the article and Appendix A.

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
