# Peer review of "A BPTF Inhibitor That Interferes with the Multidrug Resistance Pump to Sensitize Murine Triple-Negative Breast Cancer Cells to Chemotherapy"

_ijms, 2024, doi:10.3390/ijms252111346_

Round 1
Reviewer 1 Report
Comments and Suggestions for Authors
The current manuscript reported a new mechanism of action for a BPTF inhibitor, AU1, as P-glycoprotein efflux pump inhibitor. The synergistic effect with other P-gp inhibitors were found in multiple cell lines with varied P-gp expression level. The mechanism was further corroborated with known substrate P-pg in flow cytometry assay and similar trend of synergistic effect was found for another known efflux inhibitor as well. I found that the work was well-designed and supported by substantial evidence. I recommend publication after a minor revision. My other comments are listed below:
1. All the control antibody bands were under overexposure in Figure 7D and supplemental figures, and it’s difficult to quantify and normalize.
2. What’s the reason P-gp shows multiple bands on the western blots? Which band is the full-length protein.
3. Figure 10 was missing in the manuscript.
Author Response
- All the control antibody bands were under overexposure in Figure 7D and supplemental figures, and it’s difficult to quantify and normalize.
Thank you for bringing this to our attention. We have adjusted the exposures for these blots and inserted the corrected images and quantifications (see revised changes in Figure 7D on page 10, after line 347 in the main text as well as in Supplemental figure 3 in the resubmitted supplemental figures document).
- What’s the reason P-gp shows multiple bands on the western blots? Which band is the full-length protein.
This is certainly a legitimate question. Below are blots on the website of the vendor for P-Glycoprotein that are similar to what we have reported. The datasheet indicates the observed MW range to be between 130-180 kDa.
- Figure 10 was missing in the manuscript.
Thank you for pointing this out. We have no idea how or why this happened; Figure 10 is included on page 13 after line 437 in the revised manuscript.
Reviewer 2 Report
Comments and Suggestions for Authors
This manuscript presents the in vitro TNBC cell growth inhibitory effects of combining the small molecule AU1, designed as a BPTF inhibitor, with chemotherapy agents. The studies indicate that efficacy of combined agents is pronounced in TNBC cells that have high expression of P-gp and when the chemotherapy agent is a known P-gp substrate, thus providing evidence for P-gp inhibition by AU1 in TNBC cells. Review of the manuscript raised the following points and comments for the authors to consider.
Points to be addressed:
-The Results mentions Figure 10, but this figure is missing from the main manuscript (it appears Supplementary Figure 5 contains elements described for Figure 10).
Additional minor comments:
-The authors could add more discussion of previous publications using other P-gp inhibitors and their efficacy, relative to AU1, in combination with therapy against TNBC.
-Please ensure all references are cited within the main text of the manuscript (some references in the bibliography did not have matching citation numbers within the main text)
Author Response
Points to be addressed:
1.) The Results mentions Figure 10, but this figure is missing from the main manuscript (it appears Supplementary Figure 5 contains elements described for Figure 10).
Thank you for pointing this out. We have no idea how or why this happened; Figure 10 is included on page 13 after line 437 in the revised manuscript.
Additional minor comments:
1.) The authors could add more discussion of previous publications using other P-gp inhibitors and their efficacy, relative to AU1, in combination with therapy against TNBC.
We appreciate this suggestion; however, this would not be feasible within the context of the current manuscript as many P-gp inhibitors have been tested, and an entire separate manuscript would be required to cover this topic. We respectfully refer the reviewer to the following papers that address this topic at length: PMID: 33312947 and PMID: 34909680. The second paper required over 200 references to cover this topic.
2.) Please ensure all references are cited within the main text of the manuscript (some references in the bibliography did not have matching citation numbers within the main text)
Thank you for noting this error! Page 17, lines 644 through 684, have been updated and corresponding in-text references have been adjusted accordingly.
Reviewer 3 Report
Comments and Suggestions for Authors
In this manuscript, the authors report on the efficacy of AU1, a small molecule developed as an inhibitor of BPTF (bromodomain PHD finger containing transcription factor), which sensitizes preclinical models of triple-negative breast cancer (TNBC) to chemotherapy. The authors previously published that the sensitization was partly due to the promotion of autophagy. In this study, the authors identified an additional property of AU1: it appears to inhibit the P-glycoprotein efflux pump. While the presented data robustly demonstrate the efficacy of the small molecule when combined with different chemotherapies, the findings are validated only in murine TNBC models and not in any human cell lines. Additionally, the mechanistic understanding of the synergy remains incomplete. Overall, due to the limited translatability of the findings and the insufficient insight into the mechanisms behind the measured synergistic effect, the current manuscript could be significantly improved and have a greater impact if the authors address the following comments.
Major comments:
1) Combinatory Treatment in Human Cell Lines: The authors tested the combinatory treatment in MDA-MB-231 cells, which unfortunately lack the specific pump, so AU1 does not have a synergistic effect. An additional set of dose-response assessments using the MDA-MB-231 cell line overexpressing the transporter and other human TNBC cell lines with endogenous overexpression of this transporter would greatly enhance the relevance of their findings for human settings.
2) Validation of AU1 Specificity: Knockdown (KD) of MRP1 in the 4T1 cells to validate the specificity of AU1’s drug target or overexpression of this transporter to increase sensitization in MDA-MB-231 cells would serve as ideal control experiments to support the proposed mechanism of action of AU1.
3) Analysis of Cell Death and Growth Suppression: The authors demonstrated the synergistic effect inducing cell death using FACS analysis. However, they also state in the manuscript that the combinatory treatment suppresses cell growth. Additional FACS analysis of cell cycle distribution upon treatment would support this statement and show how the cells are arrested upon treatment. Furthermore, the contribution of autophagy to the observed cell death/cell growth arrest is missing. Mapping these multiple cell fate phenotypes for the different treatments would provide more insight into the possible mechanisms of action.
Minor comments:
1) Formatting/Layout: Improve the formatting/layout of both the main and supplementary figures (e.g., supp fig 1 and figure 9 / graph and titles not aligned).
2) Methods & Materials:
· Provide more detailed information on the used protocols to enhance reproducibility and repeatability. Incubation times/exposure times are not detailed enough.
· Line 114: Specify the duration for which cells were allowed to adhere and incubated with the AU1 compound.
· Explain why cells were trypsinized for the western blot assay instead of being scraped.
· Line 179: Clarify the sentence: "As appropriate, AU1 was administered overnight for 30 minutes before running the assay the following day."
· Line 190: Avoid repetition of "assay."
3) Results:
· Share data on the screen conducted on the 4T1 cells with BPTF depletion, including the efficiency of the knockdown.
· Combination therapy: Discuss the lack of significant effect of continuous exposure to AU1+vinblastine versus AU1 being removed after 4 days. Address the significant cell death induced by combinatory treatment in 4T1 cells and include data for MDA-MB-231 and E0771-LMB cell lines.
· Address the observed regrowth after 16/20 days, which may suggest a small portion of cells are arrested in their cell cycle. Additional FACS analysis to support the statement that cell growth is suppressed by both AU1 and verapamil is necessary to support the hypothesis that both inhibitors may share the same mechanism of action.
4) Discussion:
· While the findings are interesting and supported by relevant data, it should be clearly stated that AU1 has no effect on the true basal B/TNBC subtype represented by the MDA-MB-231 cell line. The scope of the application is therefore questionable. Validating the findings in a set of human TNBC cell lines displaying overexpression of MDRPs would be more translational.
· The conclusion that AU1 sensitizes TNBC cells to chemotherapy via inhibition of the P-gp efflux pump is strongly supported by the data, but it is all done in murine cell line models so the conclusions and take home message should be more nuanced.
· The contribution of autophagy is overlooked in the current study despite previous publications by the authors.
· By addressing these comments, the manuscript would greatly benefit in terms of clarity, robustness, and relevance to human cancer treatment.
Author Response
Major comments:
1) Combinatory Treatment in Human Cell Lines: The authors tested the combinatory treatment in MDA-MB-231 cells, which unfortunately lack the specific pump, so AU1 does not have a synergistic effect. An additional set of dose-response assessments using the MDA-MB-231 cell line overexpressing the transporter and other human TNBC cell lines with endogenous overexpression of this transporter would greatly enhance the relevance of their findings for human settings.
We completely agree that these types of studies would enhance the relevance of the findings for human settings, as indicated by the reviewer. However, the current work was designed solely to establish proof of concept for the capacity of the AU1 compound to modulate the function of the multidrug efflux pump, which we believe has been satisfactorily accomplished with the complementary data provided. Furthermore, as there are studies that show that the transporter pump is similar when expressed in human and murine cells (we ask the reviewer to please refer to PMID 29936088), such studies would be, at best, confirmatory, and would not alter the conclusions of this paper. Nevertheless, the reviewer raises a salient point, and we now emphasize in the Discussion section (page 15, lines 512 through 517) that whereas our data has been generated in murine breast tumor cells, P-gp is highly similar in mice and humans, and the findings are thus relevant to the human setting.
2) Validation of AU1 Specificity: Knockdown (KD) of MRP1 in the 4T1 cells to validate the specificity of AU1’s drug target or overexpression of this transporter to increase sensitization in MDA-MB-231 cells would serve as ideal control experiments to support the proposed mechanism of action of AU1.
We fully agree that these would be excellent complementary proof of concept studies. However, performing such studies would delay publication of this work by months and would not change the conclusions presented., Nevertheless, we have modified the Discussion section (page 15, lines 518 through 526) to include the potential utility of the indicated additional experimental approaches.
3) Analysis of Cell Death and Growth Suppression: The authors demonstrated the synergistic effect inducing cell death using FACS analysis. However, they also state in the manuscript that the combinatory treatment suppresses cell growth. Additional FACS analysis of cell cycle distribution upon treatment would support this statement and show how the cells are arrested upon treatment. Furthermore, the contribution of autophagy to the observed cell death/cell growth arrest is missing. Mapping these multiple cell fate phenotypes for the different treatments would provide more insight into the possible mechanisms of action.
We appreciate this thoughtful suggestion. However, we would respectfully argue that the temporal response data in Figures 5A, 5B, and 8B provide clear and unequivocal evidence of growth arrest. Furthermore, the promotion of extensive apoptosis upon combination of the antitumor drugs with AU1 (Figures 6 and 8) provides an explanation of the basis for the observed responses (i.e. that the prevention of drug efflux will result in increased accumulation of the antitumor drugs in the cells, leading to enhanced apoptotic cell death, and consequent suppression of tumor growth).
Nevertheless, the reviewer does raise an important question relating to the cell fate phenotype. To address this question, we have tested the postulate that the prolonged growth arrest may be related to senescence, as we have observed in multiple previous studies of chemotherapy and radiation effects in tumor cells from our laboratory, albeit in the absence of an efflux pump inhibitor. We now provide supplementary data (Supplementary Figure 3), indicating the promotion of senescence, as measured by the β-galactosidase probe C12FDG. We mention these findings on page 10 lines 324 through 331 in the Results section and page 14 lines 473 through 477 in the Discussion section. To our knowledge, this may be the first study to demonstrate senescence as an outcome of chemotherapy + pump inhibition in the literature, and these observations are likely to lead to similar experiments and assessments in other laboratories.
Autophagy was invoked in our previous studies in that the influence of AU1 on the multidrug resistance pump had not been identified. Rigorous studies relating to autophagy (pharmacologic and genetic inhibition) would also involve months of additional effort and, as above, would not alter the basic conclusions of this work.
Minor comments:
1) Formatting/Layout:
- Improve the formatting/layout of both the main and supplementary figures (e.g., supp fig 1 and figure 9 / graph and titles not aligned).
We modified the formatting and layout as suggested by the reviewer.
2) Methods & Materials:
- Provide more detailed information on the used protocols to enhance reproducibility and repeatability. Incubation times/exposure times are not detailed enough.
We appreciate this suggestion and have made the appropriate modifications on page 3 and throughout lines 115-141.
- Line 114: Specify the duration for which cells were allowed to adhere and incubated with the AU1 compound.
We provided the requested details of the experimental procedures on page 3 and throughout lines 115-141. We appreciate this guidance.
- Explain why cells were trypsinized for the western blot assay instead of being scraped.
This has been the standard protocol in our laboratory for many years. We altered the text to avoid any confusion on page 4, line 162.
- Line 179: Clarify the sentence: "As appropriate, AU1 was administered overnight for 30 minutes before running the assay the following day."
Thank you for bringing this oversight to our attention. We have modified the text to provide greater clarity on page 4, lines 181-182.
- Line 190: Avoid repetition of "assay."
Thank you for noting this error. We removed the duplicate wording.
3) Results:
- Share data on the screen conducted on the 4T1 cells with BPTF depletion, including the efficiency of the knockdown. Provide information.
This was clearly an oversight/omission on our part. The BPTF expression of the KD cells was shown in our previous paper (PMID: 33811160), and we now refer to this paper in the manuscript on page 5, lines 224-225. The screen to which we are referring represents a massive collection of data that evaluated over 1500 compounds of numerous drug classes; these included simple sugar alcohols, sedatives, beta blockers, anti-inflammatory compounds, anti-parasitic medicines, and known chemotherapeutics (among many others). We had used this screen as a jumping off point and explored the drug classes to which the KD cells were sensitive. However, this screening process is not actually relevant to the current work with AU1. Inclusion of the screen would actually be confusing to the readers as it would bring up the topic of epigenetic regulation, whereas the current findings involve direct interaction of the AU1 compound with the efflux pump.
- Combination therapy: Discuss the lack of significant effect of continuous exposure to AU1+vinblastine versus AU1 being removed after 4 days. Address the significant cell death induced by combinatory treatment in 4T1 cells and include data for MDA-MB-231 and E0771-LMB cell lines.
We apologize if we may have created some confusion regarding the execution of our combination treatments. In fact, AU1 + chemotherapy was not continued for the duration of the assay. For combination treatments, as indicated in the methods (and now hopefully clarified), we treated cells with the indicated cancer chemotherapeutic drug + AU1 for 96 hrs (until D4), after which we removed the chemotherapeutic drug, continuing the assay in either AU1 only (“continued”) or vehicle control only (“removed D4”). We mention that there is no difference in these groups (cont AU1 vs vehicle control) in the Results on page 9, lines 320-323.
We address the significant cell death of combination treated 4T1 cells in Figure 6, where a dramatic increase in the percentage of apoptotic cells suggests that the predominant mechanism of cell death for the combination treatments is apoptosis. As dose-response assessments for MDA-MB-231 cells demonstrated minimal IC50 fold changes between chemotherapy and chemo+AU1 groups (see Figure 3 and Table 3), we saw no need to further probe for any mechanisms of cell death. We focused on the 4T1 cell death as the IC50 fold changes between chemotherapy and chemo+AU1 groups for 4T1 cells were more pronounced than for the E0771-LMB cells.
- Address the observed regrowth after 16/20 days, which may suggest a small portion of cells are arrested in their cell cycle. Additional FACS analysis to support the statement that cell growth is suppressed by both AU1 and verapamil is necessary to support the hypothesis that both inhibitors may share the same mechanism of action.
As noted above, the regrowth of cells is definitely of further interest and we now include additional data (Supplementary Figure 3), indicating the induction of senescence in the combination-treated cells, as addressed in the reviewer’s major comment #3. We address the regrowth in the discussion (page 14, lines 473-477), referring to other instances of proliferative recovery from senescence that we have reported as a common outcome of drug-mediated growth arrest.
The reviewer is correct that without further experimentation we cannot conclusively state that both inhibitors (AU1 and verapamil) may share the same mechanism of action. We address this limitation at the end of the Discussion on page 15, lines 518-524.
4) Discussion:
- While the findings are interesting and supported by relevant data, it should be clearly stated that AU1 has no effect on the true basal B/TNBC subtype represented by the MDA-MB-231 cell line. The scope of the application is therefore questionable. Validating the findings in a set of human TNBC cell lines displaying overexpression of MDRPs would be more translational.
We believe that the relevancy of P-gp inhibition remains a valid avenue for human TNBC treatment even in the absence of additional studies involving human P-gp expressing/overexpressing cell lines. PMID 29936088 demonstrates that P-gp binding modes are similar between murine and human species. However, we appreciate the relevance of the reviewer’s comments and the necessity of highlighting these points in the Discussion (page 15, lines 514-517).
- The conclusion that AU1 sensitizes TNBC cells to chemotherapy via inhibition of the P-gp efflux pump is strongly supported by the data, but it is all done in murine cell line models so the conclusions and take home message should be more nuanced.
As indicated above, PMID 29936088 validates that P-gp binding modes are similar between murine and human species. We have addressed this point in the Discussion (page 15, lines 514-517).
- The contribution of autophagy is overlooked in the current study despite previous publications by the authors.
Apoptosis appears to be the predominant mechanism of cell death with senescence contributing to the prolonged growth arrest (senescence studies are now included in the manuscript in Supplementary Figure 3). We identified autophagy as contributing to sensitization in the previous work that focused on AU1 involvement in epigenetic regulatory pathways. This would not be relevant to the current findings. Furthermore, rigorous studies relating to autophagy (pharmacologic and genetic inhibition) would also involve months of additional effort and would not alter the conclusions of the current manuscript.
- By addressing these comments, the manuscript would greatly benefit in terms of clarity, robustness, and relevance to human cancer treatment.
Round 2
Reviewer 1 Report
Comments and Suggestions for Authors
The current revision has addressed my comments. I’d like to recommend its publication in the journal.
Reviewer 3 Report
Comments and Suggestions for Authors
While the authors have provided extensive responses to the comments I raised during my thorough review of the manuscript, they ultimately agree with the suggestions but cite time constraints and publication delays as reasons for not conducting some of the proposed experiments. This is understandable, but somewhat disappointing, as the additional experiments would have strengthened the study and increased its relevance to the human cancer context.
Despite these concerns, it is important to acknowledge that the work presented is of very good quality. The study is robustly conducted and highly relevant, particularly within the murine model using the 4T1 cell line. The innovative results obtained in this research is of interest to the scientific community.
However, the issue regarding the specificity of the Bptf inhibitor remains unresolved. The authors rely on previously published work from their lab to support their claims, which partially addresses this concern, but it does not fully eliminate it. Additionally, the absence of experimental data in human cell lines limits the study's impact in a human cancer setting. As I suggested in my initial review, critical experiments using AU1-expressing TNBC cell lines would have been highly advisable.
Regarding my comments on cell cycle arrest, the authors' response remains unsatisfactory. They argue that a reduced cell count is indicative of cell cycle arrest, but this could equally be due to increased cell death, which has not been adequately ruled out.
While the authors have acknowledged the limitations of their study in the revised discussion, the lack of critical validation experiments remains a concern. Nevertheless, given the quality and relevance of the work, I recommend that the editors make their own judgment on whether this manuscript should be accepted as it is for publication.